# Peer review of "Comparative Efficacy of First-Line Immune-Based Combination Therapies in Metastatic Renal Cell Carcinoma: A Systematic Review and Network Meta-Analysis"

_cancers, 2020, doi:10.3390/cancers12061673_

Round 1

Reviewer 1 Report

The immune checkpoint therapies are emerging as a very promising class of treatment for metastatic renal cell carcinoma. There is need for systemic reviews of related clinical data.

Authors have meticulously presented a well-organized review of data of clinicaltri-al.gov website, Pubmed, and the Cochrane library, first-line setting of patients with mRCC. They have also rightly analyzed data in the context of International Metastatic RCC Database Consortium risk classification.

Author Response

Dear Rewiewer,

On behalf of my co-authors, I would like to thank you for your valuable  comments.

Best,
R. Elaidi

Reviewer 2 Report

Basic reporting

The last decade has witnessed an unprecedented development of novel therapeutic options for metastatic renal cell carcinoma (mRCC), including immune checkpoint inhibitors (ICIs), targeted agents and tyrosine kinase inhibitors. Immune-based combinations, including nivolumab plus ipilimumab, pembrolizumab plus axitinib, and avelumab plus axitinib, currently represent the new standard of care for first line therapy of metastatic renal cell carcinoma, without any clear rationale to prefer one over the other.

The systematic review and meta-analysis by Elaidi et al. assesses the role of first-line immune-based combination treatments in mRCC.  

The manuscript is quite well written and organized.

Figures and tables are comprehensive and clear.

The introduction explains in a clear and coherent manner the background of this paper.

The statistical methods are well described. As correctly stated by the authors in the Discussion section, IMDC subgroups and geographical features could have been important confounding factors affecting the analysis. Thus, we would suggest more caution in defining the benefit of pembro-axi over the other immune-based combinations.

We suggest the following modifications:

  • With regard to the search strategy, we would recommend including the full electronic search strategy for the database used, including any limits and field tags, such that it may be replicated.

Experimental design

No comment.

Validity of the Findings

The results of the study are coherent with literature data.

The emergence of novel treatment modalities such as immune-based combination therapies has opened a new era in mRCC management. In this changing landscape, the comparison between different immune-based therapies represent an interesting and timely topic.

Lastly, we recommend including results and references related to the following paper:

  • Santoni M, Massari F, Di Nunno V, Conti A, Cimadamore A, Scarpelli M, et al. Drugs in Context 2018; 7: 212528. DOI: 10.7573/dic.212528.

General comments

We believe this article is suitable for publication in the journal although minor revisions are needed. The main strengths of this paper are that it addresses an interesting and timely question and provides a clear answer, with some limitations. We suggest including more details regarding the search strategy and the addiction of some references.

Author Response

Dear Rewiewer,

On behalf of my co-authors, I would like to thank you for your valuable  comments.

In response to your suggestions, we:

- added in background section the reference: Santoni M, Massari F, Di Nunno V, Conti A, Cimadamore A, Scarpelli M, et al. Drugs in Context 2018; 7: 212528. DOI: 10.7573/dic.212528.

- updated the appendix word file to specify the search process of our systematic review. We hope it will be explicit enough for replication.

Best,
R. Elaidi